# EFFICIENT CONTENT-BASED SPARSE ATTENTION WITH ROUTING TRANSFORMERS

## ABSTRACT

Self-attention has recently been adopted for a wide range of sequence modeling problems. Despite its effectiveness, self-attention suffers quadratic compute and memory requirements with respect to sequence length. Successful approaches to reduce this complexity focused on attention to local sliding windows or a small set of locations *independent* of content. Our work proposes to learn dynamic sparse attention patterns that avoid allocating computation and memory to attend to content unrelated to the query of interest. This work builds upon two lines of research: it combines the modeling flexibility of prior work on *content-based* sparse attention with the efficiency gains from approaches based on *local, temporal* sparse attention. Our model, the Routing Transformer, endows self-attention with a sparse routing module based on online $k$-means while reducing the overall complexity of attention to $O(n^{1.5}d)$ from $O(n^2d)$ for sequence length $n$ and hidden dimension $d$. We show that our model outperforms comparable sparse attention models on language modeling on `Wikitext-103` (15.8 vs 18.3 perplexity) as well as on image generation on `ImageNet-64` (3.43 vs 3.44 bits/dim) while using fewer self-attention layers.

## 1 INTRODUCTION

Generative models of sequences have witnessed rapid progress driven by the application of attention to neural networks. In particular, Bahdanau et al. (2014); Cho et al. (2014); Vaswani et al. (2017) relied on attention to drastically improve the state-of-the art in machine translation. Subsequent research (Radford et al., 2018; Devlin et al., 2018; Liu et al., 2019; Yang et al., 2019) demonstrated the power of self-attention in learning powerful representations of language to address several natural language processing tasks. Self-attention also brought impressive progress for generative modeling outside of language, e.g. image (Parmar et al., 2018; Menick and Kalchbrenner, 2018; Child et al., 2019) and music generation (Huang et al., 2018; Child et al., 2019).

Self-attention operates over sequences in a step-wise manner: at every time-step, attention assigns an *attention weight* to each previous input element (representation of past time-steps) and uses these weights to compute the representation of the current time-step as a weighted sum of the past input elements (Vaswani et al., 2017). Self-attention (Shaw et al., 2018) is a particular case of attention (Bahdanau et al., 2014; Chorowski et al., 2015; Luong et al., 2015).

Self-attention is commonly used in auto-regressive generative models. These models generate observations step-by-step, modeling the probability of the next symbol given the previously generated ones. At every time step, self-attentive generative models can directly focus on any part of the previous context. In contrast, recurrent neural networks (RNNs) and convolutional neural networks (CNNs) have direct interactions with only a local neighborhood of context around the current time step.

This advantage however comes at a price: unlike recurrent networks or convolution networks, the time and space complexity of self-attention is quadratic in $n$, the length of the sequence. Specifically, for every position $i \leq n$, self-attention computes weights for its whole context of length $i$, which induces a complexity of $\sum_{i \leq n} i = n(n-1)/2$. This makes it difficult to scale attention based models to modeling long sequences. However, long sequences are the norm in many domains, including music, image, speech or video generation.

Therefore, an important research direction is to investigate sparse and memory efficient forms of attention in order to scale to tasks with long sequence lengths. Previous work has proposed *data independent* or fixed sparsity patterns bounding temporal dependencies, such as local or strided attention. At each time step, the model attends only to a fix number of time steps in the past (Child et al., 2019). Extensions to local attention have suggested learning the length of the temporal sparsity for each attention module in the network (Sukhbaatar et al., 2019). These strategies draw their inspiration from RNNs and CNNs and bound their complexity by attending only to representations summarizing a *local* neighborhood of the current time step. Their attention matrices (matrices containing the attention weights for every pair of previous, current time-step) are natively sparse and requires instantiating only non-zero entries. While these approaches have achieved good results, fixing the sparsity pattern of a content based mechanism such as self-attention can limit its ability to pool in information from large contexts.

As an alternative to local attention, Correia et al. (2019) considers content-based sparsity, an approach allowing for arbitrary sparsity patterns. This formulation however does require instantiating a full dense attention matrix prior to sparsification through variants of $L_0$-sparsity or sparsemax approximations (Blondel et al., 2019).

The present work builds upon these two lines of research and proposes to retain the modeling flexibility of content-based sparse attention while leveraging the efficiency of natively sparse attention matrices. Our formulation avoids sparsemax variants and relies on clustering of attention instead. Each attention module considers a clustering of the space: the current time-step only attends to context belonging to the same cluster. In other word, the current time-step query is *routed* to a limited number of context through its cluster assignment. This strategy draws inspiration from the application of $k$-means clustering to Non-negative Matrix Factorization (NMF) (Lee and Seung, 2001; Ding et al., 2005; Kim and Park, 2008), which is relevant to the sparsification of non-negative matrices like attention matrices.

Our proposed model, Routing Transformer, combines our efficient clustered-based sparse attention with classical local attention to reach excellent performance both for language and image generation. These results are obtained without the need to maintain attention matrices larger than batch length which is the case with the segment level recurrence mechanism used in Dai et al. (2019); Sukhbaatar et al. (2019). We present experimental results on language modeling (`Wikitext-103` and `enwik-8`) and unconditional image generation (`ImageNet-64`). Routing Transformer sets new state-of-the-art while having comparable or fewer number of self-attention layers and heads, both on `Wikitext-103` (15.8 vs 18.3 perplexity) and on `ImageNet-64` (3.43 vs 3.44 bits/dim). We also report competitive results on `enwik-8` (0.99 vs 0.98 perplexity).

## 2 RELATED WORK

**Attention with Temporal Sparsity:** Research on efficient attention neural models parallels the advent of attention-based architectures. In the context of speech recognition, Jaitly et al. (2015) proposed the Neural Transducer which segments sequences in non-overlapping chunks and attention is performed in each chunk independently. Limiting attention to a fixed temporal context around the current prediction has also been explored in Chorowski et al. (2015), while Chiu and Raffel (2017) dynamically segment the sequence into variable sized-chunks.

Hierarchical attention strategies have also been explored: the model first considers which part of the inputs should be attended to before computing full attention in a contiguous neighborhood of the selected area (Gregor et al., 2015; Xu et al., 2015; Luong et al., 2015). Later, hierarchical attention has been simplified by Liu et al. (2018) that alternates coarse layers (attending to the whole sequence at a lower temporal resolution) with local layers (attending to a neighborhood of the current prediction).

This alternating strategy is also employed by Child et al. (2019), which introduces bounded and strided attention, i.e. attending to a fixed context in the past at a subsampled temporal resolution. This work formalizes such a strategy using a sparse attention formalism, showing how it relates to full attention with a specific sparsity pattern in the attention matrix. It shows that sparse attention is sufficient to get state-of-the-art results in modeling long sequences over language modeling, image generation and music generation. Sukhbaatar et al. (2019) builds upon this work and shows that is it is possible to obtain further sparsity by letting the model learn the length of the temporal context

for each attention module. This work also makes use of the attention cache introduced in Dai et al. (2019), a memory mechanism to train models over temporal contexts which extend beyond the length of the training batches.

**Attention with Content-Based Sparsity:** The above work mainly relies on two efficient ideas: attending to less elements by only considering a fixed bounded local context in the past, and attending to less elements by decreasing the temporal resolution of context. These ideas do not allow arbitrary sparsity patterns in attention matrices. Content-based sparse attention has been introduced to allow for richer patterns and more expressive models. Martins and Kreutzer (2017); Malaviya et al. (2018) propose to compute attention weights with variants of sparsemax. Correia et al. (2019) generalizes this approach to every layer in a Transformer using entmax which allows for more efficient inference. This line of work allows for learning arbitrary sparsity attention patterns from data, based on the content of the current query and past context. However, sparsity here cannot be leveraged to improve space and time complexity since sparsemax/entmax formulations require instantiating the full attention matrix prior to sparsification. This is a drawback compared to temporal sparsity approaches. Our work is motivated by bridging this gap and allows for arbitrary sparsity patterns while avoiding to instantiate non-zero entries of attention matrices.

**Sparse Computation beyond Attention:** Learning models with sparse representations/activations for saving time and computation has addressed in the past in various context. Previous work often refers to this goal as *gating* for conditional computation. Gating techniques relying on sampling and straight-through gradient estimators are common (Bengio et al., 2013; Eigen et al., 2013; Cho and Bengio, 2014). Conditional computation can also be addressed with reinforcement learning (Denoyer and Gallinari, 2014; Indurthi et al., 2019). In the domain of language modeling, a related work is the sparsely gated Mixture-of-experts (MOE) (Shazeer et al., 2017) where sparsity is induced by *experts* and a trainable gating network controls the routing strategy to each sub-network.

## 3 SELF-ATTENTIVE AUTO-REGRESSIVE SEQUENCE MODELING

Auto-regressive sequence models decompose the probability of a sequence $\mathbf{x} = (x_1, \ldots, x_n)$ as

$$p(\mathbf{x}) = \prod_{i=1}^{n} p_\theta(x_{i+1}|x_{\leq i}). \tag{1}$$

In neural models, the conditional distribution $p_\theta(x_{i+1}|x_{\leq i})$ is modeled by a neural network with learned parameters $\theta$ and these parameters are typically learned to maximize the likelihood of the training data. In particular, Transformer architectures have shown to reach state-of-the-art accuracy in several domains, including language modeling (Vaswani et al., 2017; Radford et al., 2018), image generation (Parmar et al., 2018) and music generation (Huang et al., 2018). Transformer models compose a series of attention modules. Each module refines the input representation by taking a weighted average of the representations from the previous modules.

For every module, the input representation is a sequence of $n$ vectors $\mathbf{x} = (x_1, \ldots, x_n)$ from a continuous space of dimension $d$. Thus one may actually treat the input sequence as a $n \times d$ matrix $X$. A self-attention layer operates on this representation. It first applies three linear projections,

$$Q = XW_Q, \quad K = XW_K, \quad V = XW_V, \tag{2}$$

where $Q, K$ and $V$ are referred to as *keys*, *queries* and *values*, while $W_Q, W_K, W_V$ are learned projection matrices.

The key and the query matrices determine the $n \times n$ attention matrix $A = \text{softmax}(QK^\top)$, where the softmax operator over matrices denotes that the softmax function has been applied to each row. $A$ may be interpreted as a matrix of weights in $[0, 1]$ where $A_{ij}$ denotes how much query position $i$ at the next layer must pay attention to key position $j$ at the previous layer. In the case of self-attention for auto-regressive models, queries attend only over keys from previous time-steps, i.e.

$$A = \text{softmax}\left(\text{ltr}(QK^\top)\right) \tag{3}$$

where ltr denotes the lower triangular operator. Given the attention matrix $A$, the next layer representation $X'$ is computed simply as $AV$. In summary,

$$X'_i = \sum_{j \leq i} A_{ij} V_j, \tag{4}$$

In practice, Transformer (Vaswani et al., 2017) adds several extensions to this basic self-attention mechanism. In particular, the result $X'$ of performing self-attention is scaled by $1/\sqrt{d}$. Moreover, each layer relies on multiple attention *heads*, i.e. each layer performs multiple projections onto triplet (queries, keys, values) and attention is performed for each head. The attention results from all heads are then concatenated. This strategy allows each head to specialize on different aspects of the input sequence. In addition, Transformer further processes the result of attention through a learnable non-linear transformation (multi-layer perceptron, $\mathrm{mlp}$) followed by a residual connection and a normalization step, i.e.

$$X' = \mathrm{layernorm}(X' + X) \tag{5}$$
$$X'' = \mathrm{layernorm}(\mathrm{mlp}(X') + X), \tag{6}$$

where $\mathrm{layernorm}$ denotes the parameterized normalization step from Ba et al. (2016). A full Transformer model is therefore a chain of attention modules (Eq. 6) preceded by an embedding module (learnable representation for symbols and their positions) and followed by a logistic classification module (learnable linear classifier to predict the next symbol).

Our work is interested in the application of the Transformer to long sequences, a challenging problem since space and time complexity of attention is quadratic in sequence length $n$. We describe various approaches to sparse attention including ours in the next section.

## 4 EFFICIENT CONTENT-DEPENDENT SPARSE ATTENTION

Attention-based models can be problematic for long sequences. For a sequence of length $n$, the full attention matrix $A$, as introduced in Section 3, is $n \times n$-dimensional and can be prohibitive to instantiate. This motivates sparse attention models, i.e. models relying on attention matrices which have a majority of zero entries.

For each query, a sparse attention model defines a set of keys which can be attended to. In the following, we introduce the set $S_i$ as the set of key positions that the query at position $i$ can attend to, i.e.

$$X'_i = \sum_{j \in S_i} A_{ij} V_j. \tag{7}$$

For example, classical causal self attention can attend to every key prior to the current query, which translates to $S_i = \{j \mid j < i\}$. Most previous work on attention sparsity defined such sets purely based on positions, independently of actual query and key vectors. For example, local attention (Luong et al., 2015) considers attending only to a $k$-long time window prior to the current query, $S_i = \{j \mid i - k \le j < i\}$. Child et al. (2019) propose block sparse attention where half the heads perform local attention, and half the heads perform *strided attention* given by $S_i = \{j \mid i - j \pmod{k} = 0, j < i\}$. Sukhbaatar et al. (2019) is also a variant of local attention where the cardinality of $|S_i|$ is learned from data with an $L_1$ penalty to trade-off sparsity with modeling accuracy.

These *local* attention sparsity variants are effective in practice since correlation between observations naturally decrease with time for many problems. In our experiments, we actually find that local attention is a surprisingly strong baseline in both image generation and language modeling: for e.g., a scaled up ImageTransformer (Parmar et al., 2018) gets 3.48 bits/dim compared to the 3.44 bits/dim reported in (Child et al., 2019). Similarly, scaled up versions of Transformer with local attention and the relative positional encoding scheme of Shaw et al. (2018) are able to get 19.8 perplexity on `Wikitext-103` and 1.10 bits per byte on `enwik-8`, while the state-of-the-art results using Transformer-XL (Dai et al., 2019) are 18.3 and 0.99 respectively. From an efficiency perspective, local attention is also interesting since sparsity patterns are regular, contiguous in memory and known in advance.

In this work, however, we are interested in a more generic formulation of attention sparsity and would like the sparsity pattern to be informed by the data, i.e., $\mathcal{S} = f(\mathbf{x})$. This approach has several modeling advantages: it can accommodate data without a clear ordering over observations. For temporal data, it can also discover patterns with greater sparsity if some types of queries have a longer lasting effect on future observations than others. Content-based sparse attention should however be carefully implemented if we need to avoid instantiating full attention matrices at any point in time.

For instance, Correia et al. (2019) infer sparsity from data but their formulation instantiates a full attention matrix before finding its sparse counterpart. Next section explains how a natively sparse approach can actually be devised inspired by non-negative matrix factorization (NMF).

## 4.1 CLUSTER ATTENTION WITH NON-NEGATIVE LOW RANK APPROXIMATIONS

For any given $n \times n$ matrix $A$, a low-rank non-negative approximation to it is of the form $H = FG^\top$ where $F, G \in \mathbb{R}^{n \times k}$ and $F, G \geq 0$. This factorization can be interpreted as follows: $n$ total items are routed to $k$ representatives determined by the attention matrix $F$, while each of the representative $k$ items perform full attention on the $n$ items determined by matrix $G$. Therefore, the whole attention matrix passes through a bottleneck of size $k$.

NMF studies algorithms to find such approximations (Tandon and Sra, 2010), for instance minimizing the Frobenius norm,

$$H = \arg \min_{G^\top G = I, F, G \geq 0} \left\| A - FG^\top \right\|^2. \tag{8}$$

Different algorithms have been proposed for that problem, with different trade-offs in terms of theoretical guarantees, actual accuracy and efficiency (Lee and Seung, 2001; Hoyer, 2004; Gemulla et al., 2011). In particular, $k$-means clustering (Lloyd, 1982) has been studied as a tractable approximation to non-negative low-rank matrix factorization problem (Ding et al., 2005; Kim and Park, 2008).

This relation between $k$-means and NMF motivates our work but cannot however be applied directly in our case. In particular, we want to avoid instantiating $A$ before approximating it. Furthermore, although we are interested in *low rank sparsity patterns*, our application context does not require $H$ itself to be low rank. We therefore propose a simpler strategy where $k$-means is applied to find the *routing pattern*, while the attention matrix itself remains full rank. Moreover, our approach maintains a single set of cluster centroids shared across examples, which allows for fast training and inference. We describe this strategy in the next section.

## 4.2 ROUTING ATTENTION WITH CLUSTERING

Our strategy follows the motivation we delineated in the previous section: we model sparse attention matrices with a low rank sparsity patterns relying on $k$-means clustering. Our strategy first assigns queries and keys to clusters. Then only queries and keys from the same cluster are considered for attention.

Precisely, our model projects keys $K$ and queries $Q$ into a routing matrix $R \in \mathbb{R}^{n \times d}$ as follows

$$R = [Q, K] \begin{bmatrix} W_R \\ W_R \end{bmatrix} \tag{9}$$

where $W_R$ is a fixed random orthonormal $d \times d$ routing projection matrix. The vectors of $R$ undergo $k$-means clustering in order to factorize the full attention matrix. The clustering parameters are the centroid vectors $(\mu_1, \cdots, \mu_k) \in \mathbb{R}^{k \times d}$. These parameters are model parameters shared across sequences. There are learned online along with the rest of the parameters, as delineated in Bottou and Bengio (1995). Once cluster membership for each position $i$ in the sequence is determined, we denote with $C_i$ the cluster corresponding to the routing vector $R_i$. This allows us to define our sparse attention strategy as

$$X'_i = \sum_{j \in C_i, j \leq i} A_{ij} V_j \tag{10}$$

where $C_i$ denotes the cluster of the vector $R_i$. In summary, queries are routed to keys belonging to the same cluster. Therefore, our attention sparsity pattern is of rank $k$, i.e. $FG^\top$ where $F$ and $G$ are binary matrices denoting cluster memberships of queries and keys respectively. Note that since we route both queries and keys via the routing matrix $R$, it follows that $F = G$. It is important to note that this low rank property only concerns the sparsity pattern, while the resulting attention matrix $\text{ltr}(FG^\top * A) = \text{ltr}(FF^\top * A)$ can however be of higher rank ($*$ denotes element-wise product).

As a last technical point, we work with keys and values which are unitary vectors, projecting them onto the unit ball immediately before computing them. This differentiable normalization (Ba et al.,

2016) is useful to link cluster memberships with proximity of queries and keys, as outlined below. We also assume that the max norm of $W_Q$ and $W_K$ are *close* to each other - for more details see Appendix A. This can be enforced by adding an auxiliary loss or by explicitly setting $W_Q = W_K$. Since $W_R$ is a distance preserving transform, we can write

$$\|R_i - R_j\|^2 = \|W_R(Q_i + K_i) - W_R(Q_j + K_j)\|^2 \tag{11}$$

$$\gtrapprox \|W_R\|^2 \left( \|Q_i - K_j\|^2 + \|Q_j - K_i\|^2 \right) \tag{12}$$

$$= 4 - 2 \left( Q_i^\top K_j + Q_j^\top K_i \right). \tag{13}$$

Thus, it follows that $\|R_i - R_j\| \leq \varepsilon \Rightarrow Q_i^\top K_j + Q_j^\top K_i \geq 2 - \varepsilon^2/2$. This means that, $\|R_i - R_j\| \leq \varepsilon \Rightarrow Q_i^\top K_j \geq 1 - \varepsilon^2/4$. Therefore, when two time steps $i > j$ are assigned the same cluster due to a small $\|R_i - R_j\|$ distance, it also means that their attention weight $Q_i^\top K_j$ is high. This analysis shows that our clustering routing strategy preserves large attention weights as non-zero entries.

Since, we route attention via the matrix $R$ we dub our model *Routing Transformer*. The computational complexity of this variant of sparse attention is $O(nkd + n^2d/k)$. Cluster assignments correspond to the first term, i.e. it compares $n$ routing vectors to all $k$ centroids in a space of size $d$. Query/key dot products corresponds to the second term, i.e. assuming balanced clusters, each of the $n$ queries is compared to $n/k$ in its cluster through a dot product of dimension $d$. Therefore the optimal choice of $k$ is $\sqrt{n}$ as in Child et al. (2019), thereby reducing overall memory and computational cost to $O\left(n^{1.5}d\right)$ instead of $O(n^2d)$ (Vaswani et al., 2017).

In practice, we apply regular online $k$-means to train the cluster centroids. However, in order to infer balanced routing patterns, we define the sets $C_i$ to be of equal size roughly $n/k \sim \sqrt{n}$, i.e. for every centroid $\mu_i$ we sort tokens by distance to $\mu_i$ and cluster membership is determined by this threshold (top-k). This strategy is simple and efficient. In particular, it guarantees that all clusters have the same size, which is extremely interesting in terms of computational efficiency on parallel hardware like graphic cards. As a downside, this assignment does not guarantee that each point belongs to a single cluster. In the future, we want to investigate using balanced variants of $k$-means (Malinen and Fränti, 2014) which is not common in an online setting.

## 5 EXPERIMENTS

We evaluate our sparse attention model on various generative modeling tasks including text and image generation. The following sections report our results on `Wikitext-103` (Merity et al., 2016), `enwik-8` (Mahoney, 2011), as well as `ImageNet-64`. We find that local attention is a surprisingly strong baseline and that our Routing Transformer outperforms Transformer-XL (Dai et al., 2019) and the Sparse Transformer model of (Child et al., 2019) on all tasks. In all our models, we allocate half the heads to do local attention and the other half to route attention as in Equation 10. We use the Adam optimizer (Kingma and Ba, 2014) with learning rate $2 \times 10^{-4}$ with $\beta_1 = 0.9$ and $\beta_2 = 0.98$ following the learning rate schedule described in Vaswani et al. (2017).

### 5.1 WIKITEXT-103

`Wikitext-103` (Merity et al., 2016) is a large public benchmark data-set for testing long term dependencies in word-level language models. It contains over 100 million tokens from 28K articles extracted from Wikipedia with an average of 3.6K tokens per article, which makes it a reference data-set to model long-term textual dependencies. We train a 10 layer Routing Transformer with 16 heads using the relative position encoding of Shaw et al. (2018) and with attention and ReLU dropout rate of 0.3 each. For routing attention as in Section 4.2 we choose $k = 16$ and attention window to be 256 during both training and evaluation. We describe our results in Table 2 and compare it to other recent work on sparse or recurrent attention such as Adaptive Inputs (Baevski and Auli, 2018) and TransformerXL (Dai et al., 2019) as well as a local attention with relative position encoding baseline (Huang et al., 2018). We find that local attention is a great inductive bias for sparse attention and is better than the adaptive methods proposed in Baevski and Auli (2018); Sukhbaatar et al. (2019). Moreover, our Routing Transformer model is able to get a test perplexity of 15.8 improving on the 18.3 obtained by TransformerXL (Dai et al., 2019) while having fewer self-attention layers, and without the need for segment level recurrence.

## 5.2 ENWIK-8

The `enwik-8` (Mahoney, 2011) is a data-set to benchmark text compression algorithms in the context of the Hutter prize. This data-set consists of the first 100M bytes of unprocessed Wikipedia. It is typically used to evaluate character-level language models. Similar to the prior work of Dai et al. (2019); Child et al. (2019) we use a sequence length $n = 8192$ and benchmark our results against various baselines including local attention. We train a 24 layer model with 8 attention heads with an attention and ReLU dropout rate of 0.4 each and using the relative position encoding of Shaw et al. (2018). For routing attention as in Section 4.2 we set $k = 32$ and attention window 256. We report perplexity of 0.99 like TransformerXL and Sparse Transformer, slightly under 0.98 from Adaptive Transformer. We show how samples of our model differs from Transformer with local attention in Appendix C.

## 5.3 IMAGENET $64 \times 64$

In order to evaluate the ability of our model to capture long term dependencies on a modality other than text, we report results on the ImageNet $64 \times 64$ data-set as used in Child et al. (2019). For auto-regressive image generation, this data-set consists of images of $64 \times 64 \times 3$ bytes represented as long sequences of length $12,288$ presented in raster scan, red-green-blue order. We train a 24 layer model with 16 attention heads, with half the heads performing local attention, and the other half routing attention as in Section 3. For routing attention we set $k = 8$, attention window 2048, batch size 1 and train our model for roughly 70 epochs as in (Child et al., 2019). We compare our model to a scaled-up ImageTransformer model with local attention (Parmar et al., 2018) and the SparseTransformer model of Child et al. (2019).

We find that local attention (Parmar et al., 2018) is a strong baseline for image generation, obtaining 3.48 bits/dim when scaled up to 24 layers and 16 heads, compared to later work like Sub-scale Pixel Networks (SPN) (Menick and Kalchbrenner, 2018). Our Routing Transformer model achieves a performance of 3.425 bits/dim (see Table 1) compared to the previous state-of-the-art of 3.437 bits/dim (Child et al., 2019), thereby showing the advantage of the content based sparsity formulation of Section 4.2.

| Model | Layers | Heads | Bits/dim |
|---|---|---|---|
| Glow (Kingma and Dhariwal, 2018) | - | - | 3.81 |
| PixelCNN (Van den Oord et al., 2016) | - | - | 3.57 |
| PixelSNAIL (Chen et al., 2017) | - | - | 3.52 |
| SPN (Menick and Kalchbrenner, 2018) | - | - | 3.52 |
| ImageTransformer (Parmar et al., 2018) | 24 | 16 | 3.48 |
| Sparse Transformer (Child et al., 2019) | 48 | 16 | **3.44** |
| *Routing Transformer* | 24 | 16 | **3.43** |

Table 1: Results on image generation on ImageNet $64 \times 64$ in bits/dim.

| Model | Layers | Heads | Perplexity |
|---|---|---|---|
| LSTMs (Grave et al., 2016) | - | - | 40.8 |
| QRNNs (Merity et al., 2018) | - | - | 33.0 |
| Adaptive Transformer (Sukhbaatar et al., 2019) | 36 | 8 | 20.6 |
| Local Transformer | 16 | 16 | 19.8 |
| Adaptive Input (Baevski and Auli, 2018) | 16 | 16 | 18.7 |
| TransformerXL (Dai et al., 2019) | 18 | 16 | 18.3 |
| *Routing Transformer* | 10 | 16 | **15.8** |

Table 2: Results on language modeling on `Wikitext-103` data-set. Local Transformer refers to Transformer (Vaswani et al., 2017) with relative position encoding (Shaw et al., 2018) together with local attention. Perplexity is reported on the test set.

| Model | Layers | Heads | Bits per byte |
|---|---|---|---|
| T64 (Al-Rfou et al., 2019) | 64 | 2 | 1.13 |
| Local Transformer | 24 | 8 | 1.10 |
| TransformerXL (Dai et al., 2019) | 24 | 8 | 0.99 |
| Sparse Transformer (Child et al., 2019) | 30 | 8 | 0.99 |
| Adaptive Transformer (Sukhbaatar et al., 2019) | 24 | 8 | **0.98** |
| *Routing Transformer* | 12 | 8 | 0.99 |

Table 3: Results on language modeling on `enwik-8` data-set. Local Transformer refers to Transformer (Vaswani et al., 2017) with relative position encoding (Shaw et al., 2018) together with local attention. Bits per byte (bpc) is reported on the test set.

## 6  ANALYSIS

We evaluate the difference in attention patterns between local and routed attention and compute the Jensen-Shannon divergence between local attention and routed attention for a random subset of heads in our network on the `Wikitext-103` data-set. The divergence is computed over the entire sequence length of $4096$. We average over $10$ runs and all the self-attention layers, and report means and standard deviations of the JSD in Table 4. For mean JSD per layer, see Appendix B. Note that the JSD is always non-negative and is upper-bounded by $0.6931$ when computed using the natural logarithm. We observe that the divergence between the different local heads is always very low compared to the divergence between local and routing attention heads, which is almost always very close to the upper-bound of $0.6931$. Divergence between different routing attention heads falls somewhere in between, being closer to the upper-bound. This shows that the attention distribution inferred by the routing attention of Section 4.2 is highly non-local in nature and different heads specialize in attending to very different parts of the input.

| $\mathrm{JSD}(local\|local)$ | $\mathrm{JSD}(local\|routing)$ | $\mathrm{JSD}(routing\|routing)$ |
|---|---|---|
| $0.1776 \pm 0.0649$ | $0.6044 \pm 0.0181$ | $0.4181 \pm 0.0415$ |

Table 4: Jensen-Shannon divergence between the attention distributions of a random local attention head and a random head that routes attention as in Section 3 averaged across all layers on the `Wikitext-103` data-set. We report means and standard deviations computed over $10$ runs.

## 7  CONCLUSION

Transformer models constitutes the state-of-the-art in auto-regressive generative models for sequential data. Their space-time complexity is however quadratic in sequence length, due to their attention modules. Our work proposes a sparse attention model, the Routing Transformer. It relies on content-based sparse attention motivated by non-negative matrix factorization. Compared with local attention models, it does not require fixed attention patterns but enjoys similar space-time complexity. In contrast with prior work on content-based sparse attention, it does not require computing a full attention matrix but still selects sparsity patterns based on content similarity.

Our experiments over text and image generation draw two main conclusions. First, we show that a carefully tuned local attention model establishes a strong baseline on modern benchmark, even compared to recent state-of-the-art models. Second, we show that the Routing Transformer redefines the state-of-the-art in large long sequence benchmarks of `Wikitext-103` and `ImageNet-64`, while being very close to do so on `enwik-8` as well. Our analysis also shows that routed attention modules offer complementary attention patterns when compared to local attention.

Overall, our work contributes an efficient attention mechanism that applies to the modeling of long sequences and redefines the state of the art for auto-regressive generative modeling. Our approach could prove useful in domains where the inputs are already sparse, such as 3D point clouds, social networks or protein interactions.

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

## A  MATRIX NORM ANALYSIS

In order to formally derive Equation 11, we assume that the linear projection matrices $W_Q$ and $W_K$ used to infer the queries and keys respectively are close to each other in max norm. More precisely, we assume the existence of a $\delta \geq 0$ such that $\|W_Q - W_K\|_\infty \leq \delta$. This assumption implies that for any vector $u \in \mathbb{R}^d$ it holds that:

$$uW_Q \leq uW_K + \delta\mathbf{1}\|u\|_\infty, \tag{14}$$

where the inequality is entry-wise and $\mathbf{1}$ is the vector in $\mathbb{R}^d$ with all 1's. In this case we first show that for any pair $i, j$ the queries and keys satisfy the following:

$$(Q_i - K_j)^\top (Q_j - K_i) = (X_i W_Q - X_j W_K)^\top (X_j W_Q - X_i W_K) \tag{15}$$

$$\leq (\delta\mathbf{1}\|X_i\|_\infty + (X_i - X_j)W_K)^\top(\delta\mathbf{1}\|X_j\|_\infty - (X_i - X_j)W_K) \tag{16}$$

$$\leq \delta^2\|\mathbf{1}\|^2 \max\left\{\|X_i\|_\infty, \|X_j\|_\infty\right\}^2 - \|(X_i - X_j)W_K\|^2 \tag{17}$$

Therefore, for small enough $\delta$, we get that $(Q_i - Q_j)^\top(Q_j - K_i) \lessapprox 0$ and so Equation 11 follows:

$$\|R_i - R_j\|^2 = \|W_R(Q_i + K_i) - W_R(Q_j + K_j)\|^2 \tag{18}$$

$$= \|W_R\|^2 \left(\|Q_i - K_j\|^2 + \|Q_j - K_i\|^2 - 2(Q_i - K_j)^\top(Q_j - K_i)\right) \tag{19}$$

$$\gtrapprox \|W_R\|^2 \left(\|Q_i - K_j\|^2 + \|Q_j - K_i\|^2\right). \tag{20}$$

Note that a special case of this assumption is when $W_Q = W_K$, i.e. queries and keys are shared, in which case $\delta = 0$.

## B  JENSEN-SHANNON DIVERGENCE OF ATTENTION DISTRIBUTIONS

In Table 4 we presented the Jensen-Shannon divergence between random local heads and random routing attention heads averaged across the 10 layers of the Routing Transformer model on `Wikitext-103`. Table 5 presents the mean and standard deviations of the JSD per layer instead of averaging them.

|  | JSD($local\|local$) | JSD($local\|routing$) | JSD($routing\|routing$) |
|---|---|---|---|
| layer 0 | $0.0038 \pm 0.0018$ | $0.4706 \pm 0.0319$ | $0.1579 \pm 0.0576$ |
| layer 1 | $0.3071 \pm 0.1217$ | $0.6674 \pm 0.0153$ | $0.5820 \pm 0.0104$ |
| layer 2 | $0.2164 \pm 0.0803$ | $0.5896 \pm 0.0249$ | $0.4015 \pm 0.0121$ |
| layer 3 | $0.1163 \pm 0.0336$ | $0.6047 \pm 0.0181$ | $0.4144 \pm 0.0264$ |
| layer 4 | $0.1840 \pm 0.0562$ | $0.6266 \pm 0.0062$ | $0.4191 \pm 0.0879$ |
| layer 5 | $0.2284 \pm 0.0225$ | $0.6463 \pm 0.0155$ | $0.4687 \pm 0.0449$ |
| layer 6 | $0.1901 \pm 0.0525$ | $0.6471 \pm 0.0040$ | $0.5175 \pm 0.0469$ |
| layer 7 | $0.1566 \pm 0.0685$ | $0.5798 \pm 0.0235$ | $0.4350 \pm 0.0139$ |
| layer 8 | $0.1638 \pm 0.0739$ | $0.5993 \pm 0.0148$ | $0.4268 \pm 0.0291$ |
| layer 9 | $0.2095 \pm 0.0560$ | $0.6127 \pm 0.0053$ | $0.3581 \pm 0.0019$ |

Table 5: Jensen-Shannon divergence between the attention distributions of a random local attention head and a random head that routes attention as in Section 3 per layer on the `Wikitext-103` dataset. We report means and standard deviations computed over 10 runs and use the natural logarithm so that divergences are upper-bounded by 0.6931.

## C  SAMPLES

We generate samples from the Routing Transformer model for the task of character level language modeling on the `enwik-8` data-set. We compare the generation to that from a Local Transformer model with the same number of self-attention layers and attention heads. For both the models we

generate unconditional samples using random sampling with a temperature of $1.0$. The generation from the Routing Transformer is in Table 6 while the generation from Local Transformer is in Table 7, with spelling mistakes highlighted in red. Comparing the two samples we see that the Local Transformer makes significantly more spelling mistakes, especially for long words and phrases.

Modern Least Rule to bon air and dogmatic television articles several systems: expanding
the world usually. The story differs, that would its part flex poetry will support the very little
able to put the name by oppose the stories and motorcycle transecurity and biggest life,
see the guardian article and looks extraction of large story in storage by meanching
up biggest among other items. During that time, Nevis had biggest the Very left record
of the party's tissues. The London meaning;biggest guardians of the West;
in the 1980s and 1980s. The composer Albert Director boards the capture to the beginning.
The Son Revised publicity and Revisionism board was way of to enough a descendant the President
solely changed from [[Eddie Telling]], but in fact that assembly was to become common.
The President has poor party detonation the decisions; Telling guards [[Pope University of New York]],
especially [[Canes Combustion|Canes]], and thus [[bankruptcy|bankruptcy]] student [[Advisor]]
include the latter that the swash of its populatiis churches,
not in steel [[Libertarianism peaceful anti-instance|save the matter's pieces]], and has been
languaged efforts taken in 58,000 sections of anthropic perimeter. The great
precision exists in 2004, with an assistant to feature vault on other great Peaceful themes in America,
the nose of highly [[artificial rate]]es, the discussions of cause,
simply soon because they order to setting out the institution political party activity. As of 2004,
anthropology saves them as chiefs derivation from the princess of the Executability can be run all
down by deriving the executability to understand the scientific additional British family traitirity.
These intermediate are unproperly equipped, more unprofitability the officiely competite
mass best science fiction between the notion of a brought Executive school. This project proves the
materiel of controversy and high-school intervention and thoughts of as securety.
[[Danceholding]], one can specure even as the mission to bind its dusting intervention.
At its original time, he current in the cost of moral intervention, in an 2006 slight, tracement
saw for the passagecomic belief between the city and applying its uncurrent
binary placement, one applies to specure the [[biography|biography]] of regions,
his conserved reference election in the letter the most part of the [[Dominicus]].
[[Algebraizing theory|Algebrric]] [[critic]]s, which operate the [[third type|third]]
of uncut in to the [[militity]] of the country, her brother's
deduction may not be denying cases of militity. In the period, [[Bninity Memory]]
plus how [[Jewish Philology|Philosophy]] with a series of voruments of each other can be defined
as part of [[p-cyclic memory]], which have been ruleful out
for military conventions and have unknown orders, of their proposition (because of the
political magnitude circumference of other classness and [[forgery]]).
This highly proposing not with a member of
[[John Hope (fiction)|John Hope]]: Their chief acceptance, or that rapidly proposes
C-them by bluetooth sentiments another time. The sentence of the
[[North American Executive Department|DED]] was did not believe the pocket currently by
critic, and the reaction by 2004 roughly 400
people and King John Hope was used by the capabalance of other executive
organizers in the circumstance of the chief
post-John chief of the 3000 pects. Independences improved [[North American_powers_turning_
in _the_education|north of the Education]] For throughout
the end of [[1990]]. After prizes of those of the John Hope, Generic Attempts - from London
rivers Genero helped to prevent any anti-country packaging in London. Long term helped by all the
Jury before the Council of Zoroastries develop for packet (since [[1990]]) which conspires,
any preventing them, which offered the Native American forces
to meet them up them to be referred to unto [[first Friend|southwest the first V0]].
The symbol was based on the canon of [[St. Francisco]] that bring through this force, some position
of north the Graduate Agency (which constituted a third) or the other medium in the Friend. Military as
it is clear for all the first development of one [[aar-port]], which is examined by fraud through the
[[Saujuk Earth, Illinois|Saujuk Earth]] for jewning the [[French Ten Moment]]
(2006 to 2006). At the Movement, Aar-port wanted, the city became the [[19th century]] [[Illinois]]
of Vice Presidential Aar-ranked Voyage. In the mid left, it includes a river of the Movement
through [[Nise]]. Craft dissipatable from the promiser;
Kumat e Thurberson; it includes [[Modern India]], North, and the [[Illinois Orchestra]], when three more
stronger cities neighbor than Illinois in the country. Literally, Spanish [[July 2005]] helps the city
then character spiritually ultimately northern Hinduism of [[German legislature|German]] and
[[German legislature|German]] "[[Sir John Watching the Public]]", which proves him into the port
and the Public's successes.

Table 6: Example unconditional character level text generation from the *Routing Transformer* model
trained on `enwik-8` with random sampling.

0 ]]oter sonries as refrien ritu] serm host teen serm hostul ritu'; baron, in one ritu';.
Refriendamonium rite host teen confriendamonium in the ritu's role of ritual resolution.
The ritual monitor was baron mentalists for hostilities on the rational series; this river
was dubion until the confriendamonitor, emerged with this concept into intermedies in one
witnessing. They introduced the series of not confriendamonium (very relief). They continued
to increase the time / India's power down. The ritual production to batted conway choosing
by the products of the whole choosing as happens as [[alignment]], and an aircraft happened
to the kind. As it returned to the operation of the whole producting the large small
scale axis. They were reproduced as logts (intwiting they also affected this drip to
the easily life) and our shall materials and provided a relative shallow to the motorum.
Batted chirality as to the operations of the ANCUP system material to comment adults
was bad, as an invasive shallow to be pressure that the program, the subject of ANCUP
has been celebrated, since the easily long easily more commonly functions should easily
be proved to prevent and group divisions. Batted chirality from the relative community
that the controversies came under the writing community led to the legal revolt From
the time and from a program that they had become explicitly especially in the following
year. In [[1984]], and depictions of the government by a subject were used, proved by
manticians, confering from their structure at [[Mid From Los Angeles|Los Angeles]],
disputes control and implementing the programming of Hannibal at least proposionals of
the [[dependency of Philharmos and Winds would make control of the protest actress from
much of their sources, making any process thates of [[contention]] of channing the
column to inland any time where at neardy Rube. Philharmos gives protest to any
great b(axis). Namely one of his reigns characterization of unofficial competitive
structural did attempt to win some results, myrmonisms, which have been successfully
greatly monthly. But Anamos meant that his reasons to his harmonists analog was cold
and officials success, quickly denominate a Japanese continued to bring the army various
times some full terms of the suffix of an obficial réligious religions were surcessful
older in Coloniaas and was set. One London, last names, comprising him to succedulate
to him deeper. Over the remarkable Submission was, Submission might be a mussel upon
his loyal rebuilding dynasty, a program and young by Coloniaas (erasmol), the troops law,
with their humans. Less, Miss missions and Cathar with their significance, the troops led by
the massive society of the Holy Roman Holy Mission document in the Coloniaan Committee,
hence is headby which the course of Summoniacs, "Black Writers", recognizing him
the body, Suspense), within the next member of the troops. ;ref name=Cathar Lesson
de Les Rise is derived from the 1990s. Suspense that she missed the course of the
erasmolents, part of the character in the member of the 1990s Character acted in some
of his coar, the 4 bill belonging to the eight members of the member of the 1990s.
It was threatened in 1997, and let directly affluencing in saltiming exponents within
the study emerge in the 1990s and [[canon]]s working in 1992, and roadly fought
in other sections to saltimine races. But failed to thought in the significant shot
[[Leibniz]]. (Fulham banned 1990 to 1999, when primitively [[eldest]]),
his deadly 299 primitive style of "Aghai 2001". Symmonists were silicated by
the [[Joey Forces]] and Net. Making the eldest years, he published a finite similar
racing, revealed by him a straight just greatly killed beyond a latest-sale public
bridge that two great first soundtracking, which "'Bolumeck Coal"'. "[[Mononymile]]
beginning, Dom., the head of the same racing of him of the eldest day1.
Japanese amount in the storm of the sense of Caw celebrated on Monthly appearing
instead of cross-music. When overslanding"' (which during his visual) is
raised in both the placement and which different from the storm, recent
elements, Bolumeck Winters Sense Miller which runs up raising ameth.;Caj is analogy.

Table 7: Example unconditional character level text generation from the Local Transformer model trained on enwik-8 with random sampling.

