# OpenReview forum: "Efficient Content-Based Sparse Attention with Routing Transformers"
_ICLR.cc/2020/Conference — Reject_

### Official Review · AnonReviewer2 · 2019-10-10
**Official Blind Review #2**

**Rating:** 3

**Review:**

Summary: This paper proposes a new mechanism for computing the attention scores efficiently in Transformers to avoid the quadratic computational cost in conventional Transformers (i.e., when computing the QK^T for self-attention). With a content-based "routing" technique ("group/cluster attention" may be a more accurate term to differentiate with the dynamics of Capsule Networks), the time steps of each layer form different clusters, and the self-attention mechanism is only performed within each such cluster. The authors verify their approach on very large-scale and challenging sequence benchmarks such as WikiText-103 (WT103) and enwik-8.

Pros:
+ The math notations are generally clear (e.g., dimensionality) and the paper is well-organized.
+ The paper addresses a very important (efficiency) question in Transformers, and gave a proper review to prior related works (such as Child et al.).
+ The experiments used to evaluate the model are all large-scale, high-dimensional and challenging tasks. The routing Transformer seems to be able to do very well on the WikiText-103 word-level language modeling task.
+ I especially like the analysis of the local vs. routing attention via the JSD score. It's a very interesting and useful way to compare the attention maps without visualizations (which many other works use).

--------------------------------

Cons/Questions:

1. Please number your formulas and equations.

2. Based on your definition of X' (at the end of page 3, $X_i' = \sum_{j < i}^n A_{ij} V_j$) and the subsequent Eq. (2), it seems that you did not add layer normalization and residual connection to the self-attention block. However, almost all prior works that use the Transformer architecture, such as Vaswani et al. [1], Child et al. [2] and Dai et al. [3] has two residual blocks: 1) X' = layernorm(SelfAttention(X) + X); and 2) X'' = layernorm(mlp(X') + X'). Is there any reason that you abandoned this design and used Eq. (2)? Could that have impacted the model performance? In addition, it seems you didn't scale QK^T by 1/\sqrt{d}, as did in prior works?

3. How are the k-means centroid vectors (\mu_1, ..., \mu_k) initialized? Picking the initial centers for k-means clustering has been long known as an interesting/challenging problem, especially in a high-dimensional space (which tasks like language modeling deal with). This challenge spawned works such as K-means++ [4]. I suggest the authors discussing their initialization scheme as well as the initialization's impact on the effectiveness of their approach.

4. In Section 4.2, you said "$FG^\top$ where $F$ and $G$ are binary matrices denoting cluster memberships of queries and keys respectively". However, your previous description seems to suggest that the clusters are formed based on the R matrix (with shape n x d), which contains both Q and K. Does this mean F = G^T? Or could position i in query and position i in key be in different clusters? (I think $ltr(FG^\top * A)$ would only make sense if F=G^T.) This part is confusing in the paper; please clarify.

5. The inequality at the bottom of page 5 (**feel free to clarify if I missed some assumptions from the paper**): given that $\|a-b\|^2 = (a-b)^\top(a-b) = a^\top a + b^\top b - 2a^\top b$, it's unclear in this case ($a = Q_i-K_j$ and $b = Q_j-K_i$) whether $2a^\top b$ is positive or negative (even if $Q_i, K_i$ are unit vectors). Therefore, I'm not sure why it is a >= relation here.
If this is incorrect indeed, then the subsequent analysis the authors made on \epsilon would be a bit problematic, too. (Additionally, the authors seem to suggest that $Q_i^\top K_j = Q_j^\top K_i$ in the subsequent analysis, which should not be the case, unless W_Q = W_K.) Also, just curious, how did you regularize $W_R$ to be orthonormal while training?

6. The authors acknowledge at the end of Section 4 that their method of assigning clusters "does not guarantee that each point belongs to a single cluster". This means that even with balanced cluster sizes, each cluster will have more than n/k routing vectors (i.e., time steps). This makes the $O(nkd + n^2d/k)$ bound the authors mentioned potentially inaccurate. On the other hand, without forcing such cluster balance, in the worst case, the complexity is still O(n^2d), correct?

7. The paper proposes to use $\sqrt{n}$ clustering centers, which makes sense from the perspective of minimizing total computations. However, since the cluster centroids are learned parameters, does this mean the trained routing Transformer cannot easily generalize to different sequence lengths easily? (e.g., train on sequence length 256 but test on sequence length 2000, or simply 15?) Moreover, on longer sequences, won't the sorting/clustering be increasingly inefficient?

8. What is the motivation of using half of the heads for local attention and the other half for routing attention (cf. the intro paragraph of Section 5)? Why not just use all of the heads with routing attention? Without ablations on this, it's hard to tell which part is contributing how much exactly.

9. Could you also show the # of parameters used by your model in Table 1-3 (as did in prior works)?

10. Do you observe better performance with deeper layers (which seems to be the case for the sparse Transformer [2])?

11. Except the computational concern, I still don't quite exactly see the motivation behind clustering attention. For language modeling, for instance, the time steps should be mutually dependent, and it's more like a connected, chain-like structure (where each word depends strongly on its neighbors). However, the cluster attention seems to group the words together, and there is no cross-cluster attention (i.e., the graph is broken into smaller components). In addition, the authors said the centroid parameters (\mu_1, ..., \mu_k) are shared across sequences. So in a certain sense, are they like the "anchors" in the (transformed) word embedding space? Are we then, in language modeling tasks, performing attentions among words only around these "anchors"?

12. While the theoretical complexity may be lower, how does the wall-clock time of an L-layer routing Transformer compare to an L-layer conventional Transformer (on the same sequence length, batch size)? In particular, although the routing mechanism avoids computing A, having to deal with each of the $\sqrt{n}$ clusters means that one will need to process the cluster-based attentions sequentially (and, as mentioned, you have 8 heads for local attention and 8 for routing attention, which I think are also processed in two steps?). Would this make the model actually slower on GPUs?

=====================

Minor issues that didn't impact the score:

13. In the equation at the bottom of page 3 (the definition of X_i'), shouldn't it be (j <= i) instead of (j < i), in the case of autoregressive sequence modeling? E.g., in language modeling, you do use the current word when predicting the next word.

14. At the end of Section 5.1, you said: "while having fewer [...] attention heads". I believe the released Transformer-XL model also used 16 heads? (https://github.com/kimiyoung/transformer-xl/blob/master/pytorch/run_wt103_large.sh)

15. How does the number of cluster centers influence the performance of the model? I think this would be an interesting ablation study to make.

=====================

While I'm impressed with the result the routing Transformer achieved on WikiText-103, I am not fully convinced the effectiveness of the approach (e.g., on the other tasks, the routing Transformer seems to be slightly worse than the SOTA transformer; this is less surprising, as the authors didn't change the underlying design of the Transformers). I think the paper can be improved by including more elements, such as ablative experiments and runtime benchmarking. Also, there seem to be some problems with the derivations that the authors made in the current version of the paper.


[1] https://arxiv.org/abs/1706.03762
[2] https://arxiv.org/abs/1904.10509
[3] https://arxiv.org/abs/1901.02860
[4] http://ilpubs.stanford.edu:8090/778/1/2006-13.pdf

**Experience Assessment:**

I have published in this field for several years.

**Review Assessment: Checking Correctness Of Derivations And Theory:**

I carefully checked the derivations and theory.

**Review Assessment: Checking Correctness Of Experiments:**

I carefully checked the experiments.

**Review Assessment: Thoroughness In Paper Reading:**

I read the paper at least twice and used my best judgement in assessing the paper.

---

> ### Author Response · Authors · 2019-11-15
> **Response to review**
>
> We thank Reviewer 3 for a careful reading of our paper and a thorough review and for the valuable feedback. We answer the various questions raised by the reviewer in the order in which they appear:
>
> 1- We have updated our draft to number every equation.
>
> 2- We apologize for the confusion, you are indeed correct that we missed mentioning the two residual blocks followed by layernorm and normalizing by sqrt(d). In practice, we were doing both of them but we accidentally neglected to mention it in our work. We have now updated the draft to reflect both these corrections.
>
> 3- Regarding the initialization, we use a uniform unit scaling initialization since the keys and queries are projected onto the unit ball before clustering (as explained in Sec 4.2). We are aware that initialization can be an issue for k-means and are aware of the great work of k-means++. However, note that these initialization methods assume a static set of points to cluster - while in our case the inputs to the k-means algorithm and number of points is dynamic. Extending k-means++ to these settings is left as future work.
>
> 4- You are right - thanks for pointing this out. Actually F = G and not G^T, since keys and queries are both routed using the matrix R.
>
> 5- Thanks for a careful reading of the derivation. As you point out, the inequality goes through under the assumption that the linear projections W_Q and W_K are equal. However, we note that the following weaker assumption is sufficient: that the spectral norms of W_Q and W_K are "close" to each other. While this can be enforced during optimization by adding an auxiliary loss, in practice we don't. We also enforce orthonormality of W_R by making it a fixed random orthonormal matrix and by making it non-trainable.
>
> 6- The bound of nkd + n2d/k is under the assumption that the clusters are balanced and trying more balanced versions of k-means is left open as future work. However, the way we implement it (and as is described in Sec 4.2), we introduce an additional hyperparameter which we call attention_window, and every cluster is allowed to attend to only that many items. So for every cluster center, you pick the attention_window many points closest to it and attend only to those points. This hard constraint ensures that we never overshoot the budget of attention_window. This means however that if the clustering is bad, then some points are never represented in any cluster and some points are represented in multiple clusters (in which case the representation is averaged). However, note that the clustering is informed by the end task performance - in which case it is incentivized to be a balanced clustering.
>
> 7- The centroids are learned from the distribution of representations of keys and queries. Under the assumption that the distribution of the keys and queries of the longer sequence tokens should also come from the same distribution, the learned cluster centroids during training can generalize to longer sequences. Regarding the length of receptive field, we can adjust the size of the attention window in our approach without the need for additional training. Note that both local attention and strided attention (where the sub-sampling is a function of input length) also could suffer from potentially degraded performance when the sequence length is increased.
> Regarding the inefficiency of computing clustering for long sequences - note that computing cluster memberships is simply matrix multiplication - and so should be optimized in most hardware such as GPU/TPU. In our experience the slowest operation is indexing (scatters, gathers), while sorting is relatively fast. Note that we were able to generate sequences of length up to 12k so even for practical long sequence problems this approach does scale. we do realize that data dependent sparsity, would be slower than fixed sparsity patterns although at the cost of expressivity, we hope that the future improvements in hardware with more flexible data routing capabilities could benefit our work. We also plan to explore routing contiguous blocks instead of individual positions which helps with efficiency.
>
> 8- We mix local and routing heads because there is still much to be gained from performing local attention. We also do the ablation where we use only local attention heads with everything else held fixed. What our work shows is that in some generative modeling problems there is also something to be gained from performing non-local attention in a content-dependent manner.
>
> 9- Routing transformers model for the wikitext 103 experiment has about 300M parameters. Transformer-xl has 257M parameters.

---

> ### Author Response · Authors · 2019-11-15
> **Response to review - continued**
>
> 10- Yes we do observe better performance by using more layers as in Sparse Transformer.
>
> 11- Other than computational complexity there are two other motivations. 1) memory, because keeping full attention weights on long sequences is not feasible. 2) the hope is that clustering corresponds to some meaningful segmentation of the representation space, e.g., in the case of images it makes sense to do self-attention only on pixels corresponding to its own semantic segment.
>
> 12- Yes, at the moment local attention baseline is faster than routing attention by a factor of 1.4x - 1.5x in wikitext-103 experiments and 2.3-2.5x in imagenet 64x64 experiment w.r.t. wall-clock time. Note that we cannot even compare wall clock-time to full attention for such long sequences (since it is infeasible).

---

### Official Review · AnonReviewer3 · 2019-10-10
**Official Blind Review #3**

**Rating:** 3

**Review:**

[EDIT: After reading the other reviews and discussion among reviewers, I have decided to downgrade my score. In particular, in addition to points raised by reviewer 2 there are concerns with regard to lack of ablation studies, and major clarity issues.]

This paper proposes content-based sparse attention to reduce the time/memory complexity of attention layers in Transformer networks. The method essentially boils down to keep a set of K mean vectors (which are learned/updated during training), which are used to provide clusters to be attended over. When combined with prior work on local attention (i.e. half of the heads are local attention, the other half are the newly proposed routing attention), but model is found to outperform, or be on par with, existing Transformer models despite generally being smaller.

Overall the empirical performance is quite impressive, especially on the highly-competitive Wiktext-103 dataset. The fact
However I had some detailed comments/questions:

- In equation 3, isn't A_{ij} still the output from the full (lower triangular) attention? Or do you change the softmax normalization such that it is over C_i?

- Some natural ablation studies are missing. How does the model do if it only uses routing attention? What about only local attention? Finally, what about full attention with O(n^2)? I understand some of the listed baselines are already working with local attention, but there are differences in setup that could contribute to differing performance. Therefore it would be good to ablate on these aspects, holding the other parts (layers/initialization/optimization algorithm etc.) constant.

- What is the *actual* running time/memory for local/routing/full attention layers? My guess is that the actual, rather than theoretical, difference would not be that great. It seems like the routing layer requires additional operations (i.e. online k-means) which could increase running time. Also, sorting by distance to mu_i doesn't seem very GPU-friendly.

- I found the connection to NMF somewhat tenuous, especially given the different objective (i.e. A is not fully instantiated). I feel that it would be more informative to have some of the ablation studies mentioned above instead.



**Experience Assessment:**

I have published one or two papers in this area.

**Review Assessment: Checking Correctness Of Derivations And Theory:**

I assessed the sensibility of the derivations and theory.

**Review Assessment: Checking Correctness Of Experiments:**

I assessed the sensibility of the experiments.

**Review Assessment: Thoroughness In Paper Reading:**

I read the paper at least twice and used my best judgement in assessing the paper.

---

### Official Review · AnonReviewer1 · 2019-10-24
**Official Blind Review #1**

**Rating:** 6

**Review:**

This paper proposes a novel way to increase efficiency for self-attention based sequence modeling neural networks. The proposed approach is incremental by combining content-based sparse attention with local/temporal sparse attention.

While the extension is incremental, they are able to reduce the overall complexity and achieve new state-of-the-art on wiki-text 103 dataset.

The paper is also very well written and easy to follow and understand.

One negative of the paper is section 4.1. I find the discussions around NMF to be somewhat orthogonal, especially considering the paper does not use NMF techniques for their clustering algorithm in section 4.2.
Would sparse coding in general be a good high level motivation for the proposed clustering?

It is also appreciated that the authors have released demo code for reproducibility.

**Experience Assessment:**

I have read many papers in this area.

**Review Assessment: Checking Correctness Of Derivations And Theory:**

I assessed the sensibility of the derivations and theory.

**Review Assessment: Checking Correctness Of Experiments:**

I assessed the sensibility of the experiments.

**Review Assessment: Thoroughness In Paper Reading:**

I read the paper at least twice and used my best judgement in assessing the paper.

---

### Decision · Program_Chairs · 2019-12-19

**Decision:**

Reject

**Comment:**

This paper proposes a new model, the Routing Transformer, which endows self-attention with a sparse routing module based on online k-means while reducing the overall complexity of attention from O(n^2) to O(n^1.5). The model attained very good performance on WikiText-103 (in terms of perplexity) and similar performance to baselines (published numbers) in two other tasks.

Even though the problem addressed (reducing the quadratic complexity of self-attention) is extremely relevant and the proposed approach is very intuitive and interesting, the reviewers raised some concerns, notably:
- How efficient is the proposed approach in practice. Even though the theoretical complexity is reduced, more modules were introduced (e.g., forced clustering, mix of local heads and clustering heads, sorting, etc.)
- Why is W_R fixed random? Since W_R is orthogonal, it's just a random (generalized) "rotation" (performed on the word embedding space). Does this really provide sensible "routing"?
- The experimental section can be improved to better understand the impact of the proposed method. Adding ablations, as suggested by the reviewers, would be an important part of this work.
- Not clear why the work needs to be motivated through NMF, since the proposed method uses k-means.

Unfortunately several points raised by the reviewers (except R2) were not addressed in the author rebuttal, and therefore it is not clear if some of the raised issues are fixable in camera ready time, which prevents me from recommend this paper to be accepted.

However, I *do* think the proposed approach is very interesting and has great potential, once these points are clarified. The gains obtained in WikiText-103 are promising. Therefore, I strongly encourage the authors to resubmit this paper taking into account the suggestions made by the reviewers.